# Long-Term Consequences of COVID-19 Disease Specific to Women: Exploratory Research

**DOI:** 10.3390/ijerph20010150

**Published:** 2022-12-22

**Authors:** Karolina Juszko, Patryk Szary, Justyna Mazurek, Sebastian Rutkowski, Błażej Cieślik, Joanna Szczepańska-Gieracha, Robert Gajda

**Affiliations:** 1Faculty of Physiotherapy, Wroclaw University of Health and Sport Sciences, 51-612 Wroclaw, Poland; 2University Rehabilitation Centre, Wroclaw Medical University, 50-367 Wroclaw, Poland; 3Faculty of Physical Education and Physiotherapy, Opole University of Technology, 45-758 Opole, Poland; 4Healthcare Innovation Technology Lab, IRCCS San Camillo Hospital, 30126 Venezia, Italy; 5Gajda-Med District Hospital, 06-100 Pultusk, Poland; 6Department of Kinesiology and Health Prevention, Jan Dlugosz University, 42-200 Czestochowa, Poland

**Keywords:** COVID-19, gender differences, long-term consequences of COVID-19 disease, mental health, quality of life, challenges, healthcare professionals, management

## Abstract

This study was designed to explore COVID-19 in a biopsychosocial model, taking into account the different mental and social consequences of the disease in women and men. A sociodemographic questionnaire containing anthropometric data, socioeconomic data, lifestyle data, health status before COVID-19, course of COVID-19, symptoms, and complications after COVID-19 was administered to 83 women and 64 men to investigate their mental health (MH) and quality of life (QoL). The Hospital Anxiety (HADS-A) and Depression (HADS-D) Scale, the Perceived Stress Scale (PSS-10) and the World Health Organization Quality of Life Scale Brief Version (WHOQOL-BREF) were adopted. Abnormal results in HADS-D and HADS-A were obtained in 33 (39.8%) women and 10 (15.6%) men and in 26 (31.3%) women and 14 (21.9%) men, respectively. Women experienced a lower level of QoL than men. The prolonged duration of COVID-19 symptoms was associated with increased anxiety in women during recovery. Good self-reported health before COVID-19 in women was associated with reduced QoL. Women had more symptoms of COVID-19 than men, and they experienced neurological complications more often. The presence of neurological complications in women appears to be associated with increased perceived anxiety and reduced QoL. This is an exploratory study whose results can influence future research with larger and more diverse samples.

## 1. Introduction

In its initial period, the COVID-19 disease surprised the entire medical world; however, a great deal of research was quickly produced, including about its epidemiology [1,2], spread [3], risk groups [4,5], and the most common symptoms [6] and complications [7,8]. Clinical studies have shown that the initial symptoms of COVID-19 are mainly related to viral pneumonia and usually include fever, cough, sore throat, muscle pain, and fatigue [6,9]. The first scientific publications from China and Italy indicated higher mortality rates from the pandemic among men than women [10,11]. Clinical studies have also shown that elderly people and those with pre-existing comorbidities are at a higher risk of dying from COVID-19 [5,12,13].

There are already publications that recognize the state of the COVID-19 pandemic as a traumatic health event [14] that may lead to the deterioration of mental health [14,15,16]. Research conducted in Great Britain found lower levels of subjective well-being and higher anxiety between the pre-pandemic and pandemic periods and that this was stronger among women and younger people [17]. Women’s health, including the mental health risks of the pandemic, has also been studied [18,19]. This study aimed to show COVID-19 in a biopsychosocial model that considers the mental and social consequences of the disease differently for women and men. Therefore, the main goal of this study was to determine whether there are differences in the mental state of COVID-19 survivors depending on the gender of the respondents. It was also important to determine the effect of the symptoms and any complications of the mental state and whether there are gender differences. It was hypothesized that women suffer from COVID-19 differently: their long-term consequences and health complications are different.

## 2. Materials and Methods

### 2.1. Participants and Study Setting

The study included 147 people who were referred by their general practitioner (GP) to pulmonary rehabilitation in stationary mode at the Specialist Hospital Ministry of the Interior and Administration in Glucholazy, Poland. On the day of admission, after completing all related formalities, patients filled in a sociodemographic questionnaire and psychometric scales in writing. In addition, all participants signed a written informed consent to participate in the research study.

Nearly 80% of the respondents experienced COVID-19, 7% had COVID-19 combined with pneumonia, and approximately 8% were diagnosed with COVID-19 and other respiratory diseases. They were mostly severely (62.6%) or moderately (29.9%) ill, and 55.8% required hospitalization due to COVID-19 and the need for oxygen therapy. The subjective self-reported health of most patients at the time of admission to the rehabilitation department was either average (67.3%) or poor (25.2%). Among people who underwent rehabilitation, there were also those who underwent COVID-19 asymptomatically (1.4%) or mildly (6.1%). People who considered their health condition good (7.5%) on admission also qualified for rehabilitation. The mean duration of COVID-19 disease was 4.7 (*SD* 3.06) weeks, the shortest duration of illness was 1 week, and the longest was 20 weeks.

The most common symptoms of the COVID-19 respondents were: fever (87.8%), cough (74.1%), muscle pain (73.5%), and loss of sense of smell or taste (61.2%). In addition, 50.3% of patients developed other symptoms. The percentage distribution of other symptoms is shown in Figure 1A. The most common complications after the end of COVID-19 treatment, according to the respondents, were pulmonary (85.7%), psychological (43.5%), neurological (36.7%), or cardiological (25.9%). Before catching COVID-19 disease, 46.3% of all respondents additionally suffered from chronic hypertension, 19.7% from diabetes mellitus, and other comorbidities occurred in 47.6% of people. The percentage distribution of comorbid types is shown in Figure 1B.

Self-reported health before COVID-19 disease was good in 68.0% of respondents, average in 27.9%, and bad in only 3.4% (no data—1 person). Before the disease, 70.7% of patients were physically active, with 34% of people exercising daily, 31.3% once or twice a week, and 21.8% sporadically (no data—19 people); 4.1% of the respondents admitted to currently smoking cigarettes, 37.4% had smoked in the past, and the average number of years spent smoking was 19.1 (*SD* 10.98). The sociodemographic data is presented in Table 1.

### 2.2. Outcome Measures

The following psychometric scales were used: Hospital Anxiety and Depression Scale (HADS), Perceived Stress Scale (PSS-10), and the World Health Organization Quality of Life BREF (WHOQOL-BREF). The Hospital Anxiety and Depression Scale (HADS) is a screening tool for the detection of anxiety (HADS-A) and depression (HADS-D) symptoms, consisting of 14 questions (7 statements for measuring anxiety and depression), each assessed on a four-point scale (0–3 points). The cut-off point is above 7 points for anxiety and above 7 points for depression [20].

The intensity of stress was measured using the Perceived Stress Scale (PSS-10). Subjective feelings related to one’s own life situation over the last month are assessed using 10 questions divided into two categories. The first category relates to the perception of perceived helplessness; the second category relates to the perception of the effectiveness of one’s actions. The assessment of feelings was included in a five-point scale (0—never, 4—always). A higher total score indicated a higher intensity of perceived stress [21,22].

The methodological basis for assessing quality of life was the World Health Organization Quality of Life Scale Brief Version (WHOQOL-BREF). The questionnaire assesses the perceived quality of life and general health of the respondents. Items are grouped into four domains: physical (WHO physical), psychological (WHO psychological), social (WHO social), and environmental (WHO environmental). The range of answers was included on a five-point scale (from 1 to 5 points: the higher the number of points, the better the quality of life). The assessment of quality of life in individual domains was expressed as mean values calculated in accordance with the key and guidelines presented by the authors [23].

A sociodemographic questionnaire was also used, containing anthropometric data (age, weight, and body height on the basis of which BMI was calculated), socioeconomic data (education, labor, employment, marital status, and having children) and lifestyle data (physical activity, smoking), health status before COVID-19 (self-reported health and the presence of hypertension, diabetes, other comorbidities), the course of the COVID-19 disease (main diagnosis, type of COVID-19, place of treatment, the need for oxygen therapy, and subjective assessment of health at admission to rehabilitation ward), and symptoms and complications after COVID-19. The data obtained was divided into categories, the analysis of which made it possible to answer the research questions.

### 2.3. Data Analysis

The data obtained was analyzed using Statistica v. 13.3 PL (StatSoft, Kraków, Poland). The research material was analyzed using descriptive statistics, including mean, median, standard deviation, quartile deviation, total range, and percentages. Pearson’s correlation was used to examine the relationship between continuous features such as: age, BMI, number of years worked, smoking period, duration of the disease, mental health, and quality of life. The relationship between continuous (mental health and quality of life) and categorical variables such as self-reported health before COVID-19, type of COVID-19, and self-reported health after COVID-19, was investigated using the Spearman’s rank correlation (*r*_s_). The Kruskal–Wallis ANOVA was used to determine the relationship between continuous (mental health and quality of life) and categorical variables (education, labor, employment, marital status, frequency of physical activity, and main diagnosis). The unpaired *t* test or Mann–Whitney *U* test were used to investigate the relationship between continuous (mental health and quality of life) and dichotomous variables such as having children, undertaking physical activity, current and past smoking, suffering from hypertension, diabetes, place of treatment, use of oxygen therapy, and occurrence of individual symptoms and complications. A chi-squared (*χ*^2^) test was used to determine the existence of gender differences in the categorical variables.

## 3. Results

### 3.1. Gender Comparison

#### 3.1.1. The Course of COVID-19 Disease

In total, 83 women and 64 men took part in the research; 81.9% of all women had experienced COVID-19, 6.0% had COVID-19 associated with pneumonia, and 12.1% had COVID-19 and other respiratory diseases. COVID-19 was severe in 62.7%, moderate in 31.3%, mild in 3.6%, and asymptomatic in 2.4% of the women. Less than half the women (49.4%) required hospitalization, but oxygen therapy was required in 53.0% of cases. Only 3.6% self-reported health at the time of admission to the rehabilitation department as at a good level, 67.5% at a moderate level, and 28.9% at a bad level.

In total, 76.5% of all men had experienced COVID-19, 9.4% had COVID-19 associated with pneumonia, and 14.1% had COVID-19 and other respiratory diseases. COVID-19 was severe in 62.5%, moderate in 28.1%, and mild in 9.4%, and no men referred for rehabilitation underwent COVID-19 asymptomatically. Most men (64.1%) required hospitalization, and oxygen therapy was required in 59.4% of cases. Self-reported health at the time of admission to the rehabilitation department was good in 12.5%, moderate in 67.2%, and bad in 20.3%.

The mean duration of COVID-19 disease in women was 4.4 (*SD* 2.63) weeks, and in men it was 5.0 (*SD* 3.53) weeks, with no significant difference between them (*p* = 0.30). There was no significant difference in terms of gender in the main diagnosis (*χ*^2^ = 0.79; *p* = 0.67), the type of COVID-19 (*χ*^2^ = 3.62; *p* = 0.30), the place of treatment (*χ*^2^ = 3.15; *p* = 0.08), the need to use oxygen therapy (*χ*^2^ = 0.59; *p* = 0.44), or self-reported health at the time of admission to the rehabilitation department (*χ*^2^ = 4.87; *p* = 0.09).

#### 3.1.2. Symptoms and Complications

There was no significant gender difference in terms of symptoms or complications. The exception is the significantly more frequent appearance of additional other symptoms (*p* < 0.05) and the occurrence of neurological complications in women (*p* < 0.01). Detailed data and analysis results are presented in Table 2.

#### 3.1.3. Health Status before COVID-19

Among all respondents, 42.2% of women and 51.6% of men additionally suffered from chronic hypertension, and 18.1% of women and 21.9% of men suffered from diabetes. There was no significant difference in terms of gender with regard to suffering from hypertension (*χ*^2^ = 1.34; *p* = 0.24) or diabetes (*χ*^2^ = 0.39; *p* = 0.53). Other comorbidities in the group of women occurred in 55.4% of examined patients and in the group of men in 37.5% of examined patients (*χ*^2^ = 4.65; *p* = 0.03). There were no significant differences in the types of comorbidities (*χ*^2^ = 9.52; *p* = 0.09). Self-reported health before COVID-19 disease was good in 67.5% of the surveyed women and 68.8% of the surveyed men, at an average level in 27.7% of women and 28.1% of men, and bad in only 3.6% of women and 3.4% of men (no data—1 woman). There were no statistically significant differences for either group (*χ^2^* = 0.03; *p* = 0.98).

#### 3.1.4. Lifestyle

Of the women, 71.1% declared that they had undertaken regular physical activity before their illness, and in terms of frequency, 32.5% said that they exercised every day, 34.9% did so once or twice a week, and 21.7% exercised occasionally (no data—9 women). Of the men, 70.3% undertook physical activity before their illness, of which 35.9% said that they did so every day, 26.6% did so once or twice a week, and 21.9% exercised sporadically (no data—10 men). Additionally, 6.0% of the surveyed women and 1.6% of the surveyed men admitted smoking, and 32.5% of women and 43.8% of men had smoked in the past. The mean number of smoking years for women (*n* = 27) was 18.1 (*SD* 12.14) and for men (*n* = 29)—20.0 (*SD* 9.91); the shortest smoking time reported for both women and men was 2 years, with the longest being 50 years in the group of women and 40 years in the group of men. There was no statistically significant difference in terms of gender with regard to self-reported health before COVID-19 (*χ^2^* = 0.03; *p* = 0.98), frequency of physical activity (χ^2^ = 0.01; *p* = 0.92), undertaking physical activity (*χ*^2^ = 0.85; *p* = 0.65), current cigarette smoking (χ^2^ = 1.84; *p* = 0.17), or past smoking (*χ*^2^ = 2.16; *p* = 0.14). There was no significant difference between the mean number of years of smoking for women or men (*p* = 0.39).

### 3.2. Mental Health and Quality of Life

Men and women starting post-COVID-19 rehabilitation differed significantly in terms of mental health and life satisfaction. In the study group, abnormal results in HADS-D were obtained by 39.76% of women and 15.63% of men and in the HADS-A test by 31.33% of women and 21.88% of men. Women experienced significantly higher levels of depression and stress than men and a lower level of quality of life as regards physical and mental functioning. The detailed data and the results of the statistical analysis are presented in Table 3.

### 3.3. Factors Affecting Mental Health (MH) and Quality of Life (QoL)

The analysis of the relationship between mental health and quality of life, and individual variables classified into categories, is presented below; the results that were statistically significant (*p* < 0.05) and the results of the second group for a given variable are included.

#### 3.3.1. The Relationship between the Course of COVID-19 Disease and MH and QoL

There was a correlation in both women and men between self-reported health after COVID-19 and mental health, and subjective poor self-reported health predisposed participants to an increased level of depression, anxiety, and stress during convalescence. Poor self-reported health after COVID-19 was associated with the group of women with low levels in all domains of quality of life. A low level of quality of life in the physical and mental domains was associated with poor self-reported health after COVID-19 in the group of men. There were also statistically significant correlations in the group of men between type of COVID-19 and the severity of depression symptoms and perceived intensity of stress, and the more severe the form of COVID-19, the higher the level of depression and stress experienced by men during convalescence. The results of the analysis are presented in Figure 2.

In the women’s group, there was a significant weak Pearson correlation (*r*_s_ = 0.25, *p* = 0.02) between the duration of COVID-19 and the level of anxiety experienced; the longer the disease lasted, the higher the intensity of anxiety during the recovery period. There was no such relationship in the group of men (*r*_s_ = 0.02, *p* = 0.89). Variables such as main diagnosis, place of treatment, and the need for oxygen therapy were not significantly associated with mental health and quality of life.

#### 3.3.2. The Relationship between Symptoms of COVID-19 Disease and MH and QoL

Loss of smell/taste as one of the symptoms of COVID-19 was significantly associated in women with quality of life during the convalescent period and especially with decreased social domain scores (*p* = 0.005; men: *p* = 0.84). The other symptoms studied—fever, cough, and muscle pain—were not significantly related to mental health and quality of life in either gender.

#### 3.3.3. The Relationship between the Long-Term Consequences of COVID-19 Disease and MH and QoL

Neurological complications in women were significantly associated with an increased level of perceived anxiety (*p* = 0.01; men: *p* = 0.53) and reduced quality of life during convalescence within the following domains: environmental (*p* = 0.003; men: *p* = 0.35) and social (*p* = 0.04; men: *p* = 0.83). The occurrence of cardiac complications in men was significantly related to the level of anxiety after illness (*p* = 0.01; women: *p* = 0.21) and the results for psychological quality of life during convalescence (*p* = 0.04; women: *p* = 0.85). The occurrence of pulmonary complications was not significantly associated with mental health and quality of life in either gender.

#### 3.3.4. The Relationship between Health Status before COVID-19 Disease and MH and QoL

Self-reported health before COVID-19 was significantly associated in women with reduced quality of life during the recovery period, especially in the following domains: psychological (*r*_s_ = −0.26, *p* = 0.02; men: *r*_s_ = 0.0215, *p* = 0.87), social (*r*_s_ = −0.3494, *p* = 0.001; men: *r*_s_ = −0.001, *p* = 0.99), and environmental (*r*_s_ = −0.23, *p* = 0.04; men: *r*_s_ = 0.15, *p* = 0.15). A weak negative correlation indicates that the better the women assessed their health before the disease, the lower their quality of life after the disease. A decreased quality of life for women in the physical domain was also associated with the coexistence of diabetes (*p* = 0.004; men: *p* = 0.64). The presence of other comorbidities was associated with a reduced quality of life during convalescence in the social domain, both in women (*p* = 0.03) and in men (*p* = 0.04). The presence of additional chronic diseases was also associated with an increased level of anxiety (*p* = 0.001; men: *p* = 0.75) and stress (*p* = 0.04; men: *p* = 0.89) in women. The prevalence of hypertension in the study group was not related to mental health or quality of life.

#### 3.3.5. The Relationship between Lifestyle before COVID-19 Disease and MH and QoL

Current cigarette smoking was associated among women with depression symptoms (*p* = 0.007; men: *p* = 0.54), increased anxiety (*p* = 0.004; men: *p* = 0.97), increased levels of stress (*p* = 0.02; men: *p* = 0.43), and reduced quality of life in the psychological domain (*p* = 0.01; men: *p* = 0.99). Smoking in the past and the period of smoking did not significantly correlate with mental health or quality of life in either women or men.

There was no significant relationship in either women or men in the study group between mental health and quality of life and the declared physical activity; however, in the group of men there was a significant correlation between the declared frequency of undertaking physical activity and the level of depression symptoms (*p* = 0.02; women: *p* = 0.30) and reduced quality of life in the social domain (*p* = 0.003; women: *p* = 0.31). Men who occasionally took up physical activity experienced a higher level of depression than men exercising once or twice a week (Figure 3A). Perceived quality of life in the social domain decreased in men along with a decrease in the frequency of undertaking physical activity (Figure 3B).

#### 3.3.6. The Relationship between Anthropometric Category and MH and QoL

Neither age nor body mass index (BMI) was significantly related to mental health or quality of life in the study group.

#### 3.3.7. The Relationship between Socioeconomic Category and MH and QoL

There was a significant correlation in women between a lack of children and depression symptoms (*p* = 0.03; men: *p* = 0.33) and an increased level of stress (*p* = 0.03; men: *p* = 0.37) and decreased quality of life in the psychological domain (*p* = 0.04; men: *p* = 0.42) and social domain (*p* = 0.04; men: *p* = 0.70). There was a significant weak relationship in men (*r* = −0.29, *p* = 0.03) between the number of years worked and the intensity of stress: shorter work experience was associated with a higher level of perceived stress. There was no such relationship in the group of women (*r*_s_ = −0.18, *p* = 0.12). Variables such as education, labor, employment, and marital status had no significant association with mental health and quality of life.

## 4. Discussion

The physical and the mental states are inextricably linked, and their separation can lead to health inequalities and a lack of appropriate holistic care, especially for the most vulnerable patients. Poor mental health results in more frequent hospitalization for somatic diseases [24,25]. On the other hand, the physical condition plays an important role in perceived well-being [26,27]; for example, depression is two to three times more common in people with chronic physical illnesses than in people who have good physical health [28].

This study was designed to demonstrate COVID-19 disease in a biopsychosocial model, taking the mental and social consequences of the disease differently into account for women and men. Women are far more likely to be diagnosed with disorders related to mental health [29,30,31]. This is affected by many factors, including sociological and cultural [31]. The main aim was therefore to investigate whether there were differences between the mental state of COVID-19 survivors according to the gender of the subjects.

Our results indicate that women experienced significantly higher levels of depression and anxiety and lower levels of quality of life in the physical and mental domains during recuperation compared to men. In the study group, 39.76% of women and 15.63% of men obtained abnormal scores on the HADS-D, and 31.33% of women and 21.88% of men on the HADS-A. The women’s scores obtained in this study were significantly higher than those obtained in other studies. Huang et al. examined the 6-month consequences of COVID-19 in patients discharged from hospital, and only 28% of women reported experiencing anxiety or depression [32]. The men’s results in our study, however, fit the pattern of results obtained in other research. In the study by Huang et al., 18% of men reported anxiety or depression [32]. In the study by Rass et al., 11% of patients reported depressive symptoms 3 months after COVID-19, and 25% reported anxiety symptoms; however, the authors did not report results by gender [33]. A meta-analysis by Rogers et al. showed that the incidence of depression was 14.9% (12.1–18.2) in the post-COVID-19 period, and for anxiety disorders it was 14.8% (11.1–19.4), although again the results were given without gender division [34].

The prolonged duration of symptoms of COVID-19 disease was associated in this study with increased anxiety in women during recovery. Self-reported health before COVID-19 in women was also significantly associated with reduced quality of life, especially in the psychological, social, and environmental domains. The better the women assessed their health before the disease, the lower their post-disease quality of life. This may have been related to the social roles performed by women. The long period of inability to perform daily duties caused them to experience heightened emotional tension, and a sense of not fully recovering from the illness was weighing on their quality of life [31]. Problems due to prolonged recovery and its consequences, including for mental health, have also been noted by other researchers [35,36]. Other studies have also observed that long COVID-19 syndrome occurs more often in women than in men [37,38,39], and Davido et. al. determined that females are four times more susceptible to long-term consequences than males [40].

The second part of this study importantly identified the effect of symptoms and complications on people’s mental states and whether there were gender differences. It turned out that the structure of presenting symptoms and long-term consequences differed between men and women. Women had significantly more symptoms of COVID-19 disease than men, and they experienced neurological complications significantly more often. Interestingly, loss of smell/taste significantly affected quality of life in the women’s group and especially reduced social domain scores. The presence of neurological complications appeared to be significantly associated in women with increased levels of perceived anxiety and reduced quality of life in the environmental and social domains. Most studies to date have focused on determining the type of symptoms and complications of COVID-19 disease without correlating their prevalence with gender [32,41,42,43,44,45] or have concentrated only on gender differences without looking for a link to mental health [46]. 

Depression and anxiety have been included in most studies, but the relationship between mental health and quality of life and other symptoms and long-term consequences has not been examined, nor has the relationship with gender. Epidemiological studies have consistently reported gender differences in mental health [47]. In general, women show higher prevalence rates of anxiety and stress disorders than men. Depressive and anxiety symptoms are the most common indicators of psychological distress [47]. Anxiety disorder has been reported to occur three times more frequently in women than men during the COVID-19 pandemic [48]. Moreover, gender is shown to be the strongest predictor of PTSS during an epidemic [49]. Thus, the higher levels of anxiety and depression and the fear of COVID-19 in women in this study were in line with previous studies showing that psychiatric effects during the pandemic have a greater impact on women [50].

It was concluded that women suffer differently from COVID-19; that is, the long-term consequences and health complications are different in women. Factors affecting the mental health and quality of life of men and women were also different, with each gender oriented toward different values. The frequency of physical activity was found to be a factor significantly affecting mental health in men. Men who sporadically engaged in physical activity before COVID-19 were characterized during the recovery period by higher levels of depression symptoms compared to the rest of the male group. Additionally, as the frequency of exercise decreased, the men’s perceived quality of life in the social domain decreased. Perhaps they realized that they would also not quickly return to their pre-disease state of health due to their previous poor motivation to undertake physical activity. Other studies show that complications after COVID-19 significantly affect the intensity of physical activity undertaken [51,52].

### Strengths, Limitations, and Future Research Direction

The main strength of this study is a new look at the symptoms and long-term consequences of COVID-19 from the perspective of their gendered effect on mental health. Nevertheless, several limitations of this study need to be acknowledged. First, the study group, due to its ethnic homogeneity (only Poles participated in the study) and small sample size, although relatively large for our conditions, is not representative of the general population. Secondly, the study only included people who were referred by their GP for pulmonary rehabilitation in stationary mode, and therefore, these conclusions should not be extrapolated to patients who do not require pulmonary rehabilitation. Thirdly, we are unable to determine the evolution and duration of all symptoms from the time of hospital discharge. Fourth, there are several psychological (e.g., post-traumatic stress), social (e.g., isolation, stigmatization), or familial (e.g., infection or death of a family member) stressors that could affect the occurrence of relationships between some of the symptoms and the long-term consequences of COVID-19 and mental health. Finally, we mostly collected self-reported patient outcomes rather than objective measures, which could help to further identify gender differences in future studies.

The results prompted us to undertake educational and mental health activities. The study showed that post-COVID-19 rehabilitation should also include psychological support therapy, especially for women. The results encourage prevention and mental health promotion programs for people who have survived COVID-19. Such programs should include pro-health education, relaxation, learning the ability to actively cope with stress. The research results show that they can be conducted both in direct contact and remotely [53].

As our study is an exploratory one, the results can influence future studies that have larger and more diverse samples as well as pre-defined hypotheses and analyses.

## 5. Conclusions

The findings of the study suggest that health-related and psychological support interventions may be more effective if they are tailored on specific sociodemographic groups. Psychological support programs specifically designed for women during the COVID-19 pandemic may provide positive improvements in terms of their stress and anxiety. However, more research is needed to provide detailed recommendations to policymakers to pursue these goals.

The results indicate that the treatment and recovery period for COVID-19 disease should be different for women and men. Overall, we believe patients reporting to their general practitioner with long-COVID-19 syndrome, especially women, should be screened for mood disorders. Any treatment based on incomplete diagnosis is not very effective. The relationship between physical and mental health is bidirectional, which is why it is so important to work hand in hand with other specialists, including a psychologist, pulmonologist, neurologist, and physical medicine and rehabilitation specialists.

## Figures and Tables

**Figure 1 ijerph-20-00150-f001:**
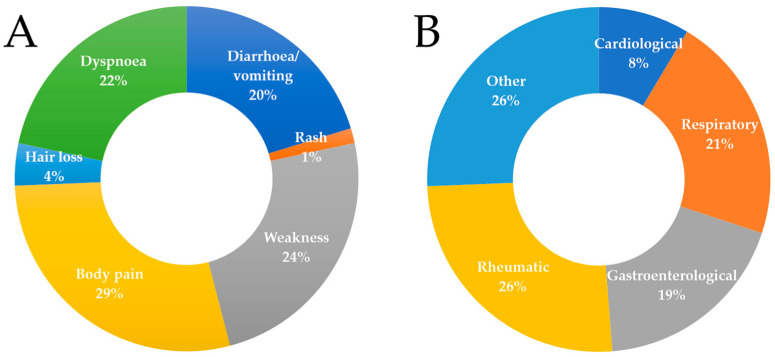
Symptoms associated with COVID-19 (**A**) and comorbidities present in the study group (**B**).

**Figure 2 ijerph-20-00150-f002:**
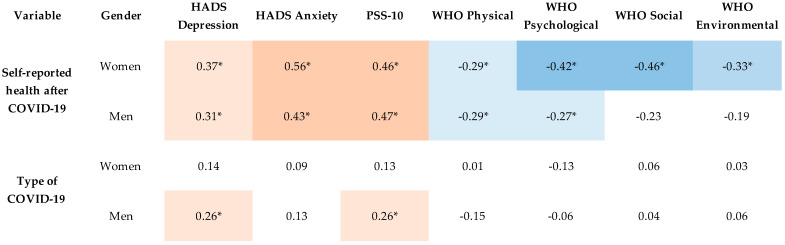
Spearman’s correlation heatmap. Significant correlation marked with asterisk. *Note*: Self-reported health was coded hierarchically (good/fair/poor), and type of COVID-19 was coded hierarchically (asymptomatic/mild/moderate/severe).

**Figure 3 ijerph-20-00150-f003:**
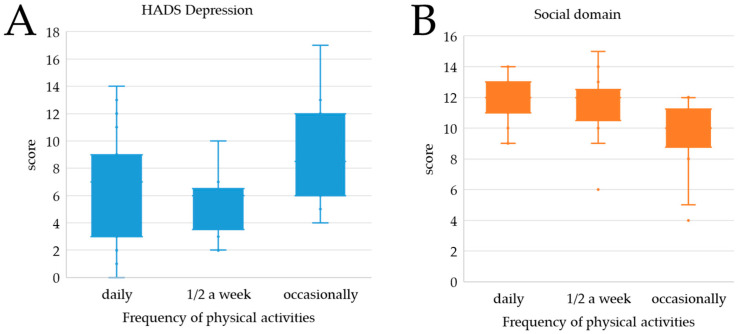
Relationship between depression (**A**), social domain (**B**), and physical activity in the group of men.

**Table 1 ijerph-20-00150-t001:** Sociodemographic data.

Variable	Total (*n* = 147)	Women (*n* = 83)	Men (*n* = 64)	*p*
Age, years, mean (*SD*)	56.0 (10.45)	56.1(9.84)	55.8 (11.28)	0.89 ^a^
Body mass, kg, mean (*SD*)	83.7 (17.07)	77.2 (14.09)	92.3 (16.96)	<0.001 ^b^
Height, cm, mean (*SD*)	168.8 (8.96)	163.5 (5.37)	175.9 (7.78)	<0.001 ^b^
BMI, mean (*SD*)	29.3 (5.12)	28.9 (5.20)	29.8 (5.01)	0.28 ^b^
Years worked, mean (*SD*)	31.1 (9.08)	30.69 (8.71)	31.6 (9.59)	0.31 ^b^
Education, *n* (%)				
Vocational	17 (11.6)	6 (7.2)	11 (17.2)	0.18 ^c^
Secondary	59 (40.1)	33 (39.8)	26 (40.6)	
Higher	68 (46.3)	41 (49.4)	27 (42.2)	
No data	3 (2.0)	3 (3.6)	0 (0.0)	
Labor, *n* (%)				
Physical and mental	39 (26.5)	22 (26.5)	17 (26.6)	<0.001 ^c^
Physical	36 (24.5)	9 (10.8)	27 (42.2)	
Mental	51 (34.7)	37 (44.6)	14 (21.9)	
No data	21 (14.3)	15 (18.1)	6 (9.4)	
Marital status, *n* (%)				
Married	104 (70.7)	50 (60.2)	54 (84.4)	<0.001 ^c^
Single	12 (8.2)	9 (10.8)	3 (4.7)	
Divorced	15 (10.2)	12 (14.5)	3 (4.7)	
Widowed	11 (7.5)	10 (12.0)	1 (1.5)	
No data	5 (3.4)	2 (2.4)	3 (4.7)	
Children, *n* (%)				
Yes	127 (86.4)	72 (86.7)	55 (85.9)	0.84 ^c^
No	13 (8.8)	7 (8.4)	6 (9.4)	
No data	7 (4.8)	4 (4.8)	3 (4.7)	
Employment, *n* (%)				
Employed	96 (65.3)	50 (60.2)	46 (71.9)	0.31 ^c^
Retired	42 (28.5)	27 (32.5)	15 (23.4)	
Pensioned	6 (4.1)	5 (6.0)	1 (1.6)	
Unemployed	2 (1.4)	1 (1.2)	1 (1.6)	
No data	1 (0.7)	0 (0.0)	1 (1.6)	

*SD*: standard deviation; ^a^ according to unpaired *t* test; ^b^ according to Mann–Whitney *U* test; ^c^ according to chi-squared test.

**Table 2 ijerph-20-00150-t002:** Percentage of incidence of symptoms and complications by gender, including statistical analyses.

	Women (*n* = 83)	Men (*n* = 64)	*χ* ^2^	*p*
Symptoms (%)				
Fever	87.9	87.5	0.01	0.93
Cough	77.9	70.3	0.87	0.35
Muscle pain	74.7	71.9	0.15	0.70
Loss of smell/taste	66.3	54.7	2.04	0.15
Other symptoms	60.2	37.5	4.65	0.03
**Complications (%)**				
Pulmonological	86.7	84.4	0.17	0.68
Cardiological	25.3	26.6	0.03	0.86
Neurological	48.2	21.9	10.77	0.001
Psychological	49.4	35.9	2.66	0.10

*p*-value as a result of chi-squared test.

**Table 3 ijerph-20-00150-t003:** Gender differences in perceived levels of depression, anxiety, stress, and quality of life.

	Women (*n* = 83)			Men (*n* = 64)			
Variable	Mean (*SD*)	Range	Me (*QD*)	Mean (*SD*)	Range	Me (*QD*)	*p*
HADS Depression	9.0 (4.48)	0–18	9 (3.5)	6.8 (3.66)	0–17	6.5 (2.5)	0.001
HADS Anxiety	8.1 (4.55)	1–18	8 (4.0)	6.9 (4.26)	0–17	7 (3.5)	0.12
PSS-10	21.1 (7.35)	2–34	22 (5.5)	17.0 (6.36)	2–29	16 (4.3)	<0.001
WHO Physical	20.6 (3.90)	14–39	21 (3.0)	21.8 (2.77)	16–28	21.5 (2)	0.02
WHO Psychological	19.8 (3.13)	12–26	20 (2.0)	20.8 (2.88)	10–26	21 (2.0)	0.04
WHO Social	11.2 (2.18)	4–15	12 (1.5)	11.2 (2.07)	4–15	12.(1.0)	0.87
WHO Environmental	28.3 (4.56)	20–37	28 (3.5)	29.0 (4.45)	11–38	29 (2.8)	0.25

Me: median; *SD*: standard deviation; *QD:* quartile deviation; HADS: Hospital Anxiety and Depression Scale; PSS-10: Perceived Stress Scale; WHO: World Health Organization Quality of Life BREF; *p*-value as a result of Mann–Whitney *U* test.

## Data Availability

Data are available from the corresponding author upon reasonable request.

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
