# Peer review of "Long-Term Consequences of COVID-19 Disease Specific to Women: Exploratory Research"

_ijerph, 2022, doi:10.3390/ijerph20010150_

Round 1
Reviewer 1 Report
Dear authors,
I applaud your efforts to investigate gender differences in the consequences of COVID-19 infection because this is an important topic that could influence clinical practice.
Your paper is easy to read and the results seem to point to some interesting conclusions.
Given the limited sample size and a lack of pre-registration with pre-defined hypotheses based on a clear theoretical framework, it seems too early to suggest different treatment and recovery trajectories based on gender.
By having so many scales and examining correlations among them, you are bound to find some significant results that look interesting. However, that is not enough evidence to influence clinical practice.
For these reasons, I see this study as exploratory and the results could only influence future studies that will have larger and more diverse samples as well as pre-defined hypotheses and analyses.
Some conclusions are very weak because they are based on subgroup analyses and include very few participants and they should all be removed.
For example, on page 10, claiming poorer mental health in women who have no children and connecting that to fertility concerns should be removed because:
a) conclusion is based on only 6 child-free women out of 83 women in your total sample
b) average age of women in the sample is 56.1 so most of them are in menopause and there are no fertility concerns
Finally, I notice a typo in figure 2. - psychological QL is written twice, while social QL is missing.
Therefore, my recommendation is to revise this paper and clearly present it as an exploratory study without suggesting clinical implications. This should be clear in the abstract, and ideally already from the title.
Best of luck with your future work
Author Response
Thank you for your very careful review of our paper, and for the comments, corrections and suggestions that ensued. A major revision of the paper has been carried out to take all of them into account. And in the process, we believe the paper has been significantly improved.
In the present „Response Letter“ we first detail the major changes that have been made in the paper to correct the main weaknesses identified by the reviewer. We then sequentially address all of the points that we „step-by-step“ corrected.

Reviewer 2 Report
The current paper, by adopting a biopsychosocial model that considers the mental and social consequences of the disease differently for women and men, aims to indagate whether there are differences in the mental state of COVID-19 survivors depending on the gender of the respondents.
It was also important to determine the effect of the symptoms and any complications on the mental state and whether there are gender differences. It was hypothesized that women suffer from COVID-19 differently: their long-term consequences and health complications differ.
The research is exciting and undoubtedly current. Overall, the paper is well-written, and recent literature is properly quoted in the introduction. However, my doubts concern the low number of research participants and especially an analysis of the data that only present simple correlations. The research could have gained more methodological robustness by analyzing the data using regression statistics, making some predictor variables explicit.
However, after my reading, I would suggest a few improvements as follows:
1. The study included 147 people referred by their primary care physician to pulmonary rehabilitation in the inpatient mode at the Ministry of Interior and Administration's specialist hospital in Glucholazy, Poland. But it needs to be made clear how in reality, they were contacted and how they answered the questions, whether through an online questionnaire or by the presence at the hospital.
2. Respect No difference in psychological symptoms. Comment on this aspect since gender difference usually affects a type of psychological symptomatology generally associated with women.
3. Concerning lifestyle, it needs to be clarified in the article what additional information the pre-infection lifestyle could give.
4. Comment on the results of Gender differences in perceived levels of depression, anxiety, stress, and quality of life and also about the absence of differences in psychological symptoms.
5. You should include a table that consists of all correlations. Also, I would recommend including more explanatory labels in Figure 2. A generic "self-reported health after Covid" or "Type of COVID-19" does not make the presentation of results precise.
6. On line 377, The authors state: "The main strength of this study is a new look at the symptoms and long-term consequences of COVID-19 from the perspective of their gendered effect on mental health". Comment in application terms on what this result entails.
7. On line 392, The authors say: "The results prompted us to undertake educational and mental health activities" Describe, for example, what educational activities.
Author Response

(The authors gave the same response as above.)

Round 2
Reviewer 2 Report
I believe the paper has been significantly improved.
Author Response
Thank you for your comments.